# How Genetic Variants in Children with Familial Hypercholesterolemia Not Only Guide Detection, but Also Treatment

**DOI:** 10.3390/genes14030669

**Published:** 2023-03-07

**Authors:** Sibbeliene E. van den Bosch, Willemijn E. Corpeleijn, Barbara A. Hutten, Albert Wiegman

**Affiliations:** 1Department of Pediatrics, Amsterdam Cardiovascular Sciences, Amsterdam Gastroenterology Endocrinology Metabolism, Amsterdam University Medical Center, Location AMC, 1105 AZ Amsterdam, The Netherlands; 2Department of Epidemiology and Data Science, Amsterdam Cardiovascular Sciences, Amsterdam University Medical Center, Location AMC, 1105 AZ Amsterdam, The Netherlands

**Keywords:** familial hypercholesterolemia, cholesterol, lipids, genetic screening, lipid-lowering therapy

## Abstract

Familial hypercholesterolemia (FH) is a hereditary disorder that causes severely elevated low-density lipoprotein (LDL-C) levels, which leads to an increased risk for premature cardiovascular disease. A variety of genetic variants can cause FH, namely variants in the genes for the LDL receptor (*LDLR*), apolipoprotein B (*APOB*), proprotein convertase subtilisin/kexin type 9 (*PCSK9*), and/or LDL-receptor adaptor protein 1 (*LDLRAP1*). Variants can exist in a heterozygous form (HeFH) or the more severe homozygous form (HoFH). If affected individuals are diagnosed early (through screening), they benefit tremendously from early initiation of lipid-lowering therapy, such as statins, and cardiovascular imaging to detect possible atherosclerosis. Over the last years, due to intensive research on the genetic basis of LDL-C metabolism, novel, promising therapies have been developed to reduce LDL-C levels and subsequently reduce cardiovascular risk. Results from studies on therapies focused on inhibiting PCSK9, a protein responsible for degradation of the LDLR, are impressive. As the effect of PCSK9 inhibitors (PCSK9-i) is dependent of residual LDLR activity, this medication is less potent in patients without functional LDLR (e.g., null/null variant). Novel therapies that are expected to become available in the near future focused on inhibition of another major regulatory protein in lipid metabolism (angiopoietin-like 3 (ANGPTL3)) might dramatically reduce the frequency of apheresis in children with HoFH, independently of their residual LDLR. At present, another independent risk factor for premature cardiovascular disease, elevated levels of lipoprotein(a) (Lp(a)), cannot be effectively treated with medication. Further understanding of the genetic basis of Lp(a) metabolism, however, offers a possibility for the development of novel therapies.

## 1. Introduction

Familial hypercholesterolemia (FH) is the most common genetic metabolic disorder in the world. Based on estimated prevalence, every minute, a baby with FH is born worldwide [1]. This semi-dominant autosomal disorder leads to severely elevated low-density lipoprotein cholesterol (LDL-C) levels, which is an important risk factor for the development of atherosclerotic cardiovascular disease (ASCVD) [2]. When left untreated, atherogenesis already starts in childhood [3]. In this review, we will discuss the importance of cascade screening and genetic testing of individuals at risk. Furthermore, we will address therapeutic developments that specifically target genetic variants causing FH. Subsequently, the role of increased lipoprotein(a) (Lp(a)) as a possible risk factor for cardiovascular disease, particularly in combination with FH, will be discussed.

## 2. Prevalence and Screening

Accurate data on the prevalence of FH in most countries are lacking, especially from non-Western countries. However, in 2020, two independent groups simultaneously published their meta-analyses of ±11 million and ±7 million individuals, respectively, from all parts of the world, showing a similar estimated heterozygous FH (HeFH) prevalence of 1:311 and 1:313 individuals in the general population [4,5]. Additionally, they identified a tenfold prevalence of FH in adults with ischemic heart disease (IHD) and even a twentyfold prevalence of FH in adults with premature IHD compared to the general population [4]. The estimated prevalence of homozygous FH (HoFH) is ~1:350.000 to 1:400.000 (calculated by 1/312 × 1/312 × 1/4) individuals. Since effective lipid-lowering treatment is available, detection—and subsequent adequate treatment—of patients with FH is essential. In recent studies, European Atherosclerosis Society (EAS) Familial Hypercholesterolemia Studies Collaboration (FHSC) showed that FH is diagnosed often later in life (median age (IQR) 44.4 years (32.5–56.5)) and undertreatment is common [6,7]. To trace affected individuals, cascade screening (genetic testing of biologic relatives after an FH index case has been diagnosed) has been performed with great success but only in some countries. In the Netherlands, over 28,000 patients with FH were found thanks to a successful national cascade screening program (active during the period 1994–2014) [8]. A major health-economic analysis in Australia showed that FH screening at the age of 10, and initiating treatment if diagnosed, resulted in 7.77 life-years gained (LYG) and 7.53 quality-adjusted life years (QALYs) per person screened [9]. Due to the nature of FH (the chance of a first-degree relative being affected by the disease in case of HeFH is 50%; in case of HoFH the chance is close to 100%), this disease would also be an excellent candidate for the implementation of a national screening program in children and a subsequent reverse cascade screening to detect the affected parent(s) [10,11,12,13]. At this moment, the majority of FH cases are diagnosed too late, when a person already has symptomatic ASCVD [7]. Even then, opportunistic screening of their relatives is important. Because therapy is preferably initiated in childhood, a consensus paper from 2015 advised that children suspected of having HeFH should be genetically screened from the age of 5 years onwards, while children suspected of HoFH should be screened as early as possible (Figure 1) [1,14].

## 3. Pathophysiology

Persistently high LDL-C levels lead to increased levels of oxidized LDL-C in the arterial wall, which initiate an inflammatory cascade, which, in turn, results in vascular damage and the formation of atherosclerotic plaques [15,16]. A large variety of genetic variants (>1700) in the cholesterol pathway can result in increased LDL-C levels [17]. Patients are genetically diagnosed as HeFH if one disease-causing variant is found in one allele, while the rare and much more severe HoFH is diagnosed if disease-causing variants are found in both alleles. The majority of genetically confirmed FH patients (95%) have a variant in the LDL-receptor (*LDLR*) gene. Less frequent forms of FH include variants in the apolipoprotein B (*APOB*) gene (2–5% of FH cases) and in the proprotein convertase subtilisin/kexin type 9 (*PCSK9*) gene (<1% of FH cases) [18]. Pathogenic variants result in failure to synthesize a functional protein or result in synthesis of a completely inactive protein, called a negative or null allele. If pathogenic variants result in a partially inactive protein, this is called a defective allele [19]. Variants in all of these genes result in a decrease in the cellular absorption of LDL-C, which, in turn, causes increased LDL-C levels in plasma. Normal LDL-C levels may differ slightly between different populations. In FH, different pathogenic variants are associated with different levels of LDL-C [20]. Patients with an *LDLR*-null allele present with higher average LDL-C levels than those with an LDLR-defective allele (6.0 versus 4.9 mmol/L; *p* < 0.001) and the more negative alleles in HoFH patients, the higher the LDL-C levels expected [21]. PCSK9 was discovered in 2003 by Seidah et al. and is formed in the endoplasmic reticulum (ER) of hepatic cells [22]. PCSK9 prevents recycling of LDLR and promotes its destruction both intra- and extracellularly. In the intracellular pathway, PCSK9 binds to LDLR at the trans-Golgi network, where destruction of LDL-C by lysosomes starts immediately. In the extracellular pathway, PCSK9 binds to epidermal growth-factor-like A (EGFA), which inhibits the binding of LDLR to LDL-C [23]. Gain-of-function variants in the PCSK9 gene (*PCSK9*) are associated with hypercholesterolemia, and loss-of-function variants in this gene are associated with hypocholesterolemia and have a more favorable cardiovascular risk profile [24]. This discovery led to the development of new therapies that are based on inhibition of PCSK9 (PCSK9-i). Due to the nature of PCSK9, the effect of inhibition depends on the residual LDLR function and is, therefore, less effective in HoFH individuals with null variants [25]. Other loss-of-function variants, namely in the gene encoding for ANGPTL3 (*ANGPTL3*), are linked to lower total cholesterol, LDL-C, and triglyceride (TG) levels [26]. Angiopoietin-like 3 (ANGPTL3) protein is secreted by liver cells and is a major regulatory protein in lipid metabolism. ANGPTL3 is an inhibitor of the lipoprotein lipase (LPL) enzyme and endothelial lipase (EL) enzyme in tissues. Results of a large human genetics study consisting of 58,335 participants (13,102 patients with coronary artery disease (CAD) and 40,430 controls) reported that *ANGPTL3* loss-of-function variants were associated with 41% lower odds of CAD (adjusted OR 0.59, 95% CI 0.41–0.85; *p* = 0.004) [27]. Novel therapies that are focused on inhibition of ANGPTL3 production are effective independently of the residual LDLR function. In addition to gene-based therapy, new developments in cellular-based therapy might be effective in the future treatment of FH [28,29].

## 4. Diagnosis

Aside from the arcus cornealis, xanthomata, and xanthelasmata, FH is an asymptomatic disease until a person has cardiovascular complaints [30]. Therefore, the (timely) diagnosis of FH has important consequences for an individual and likely for first-degree family members. After being diagnosed with FH, individuals should be strongly encouraged to optimize their lifestyle by exercising, eating healthy food, and either to not commence or to stop smoking. Nevertheless, to normalize life expectancy, the lifelong use of lipid-lowering medication is often unavoidable. There are different diagnostic tools available to diagnose FH, using patient’s characteristics, family history, and lipid levels. The Dutch Lipid Clinics Network criteria (DLCN) are not valid for children, but the Simon Broome criteria and the US (MEDPED) diagnostic criteria for FH do have specific cut-offs for LDL-C levels in children. When genetic confirmation of FH, being the gold standard, is not possible, clinical criteria for both HeFH and HoFH are available and validated (Figure 1) [1,18]. Genetic confirmation of FH can be either performed through Sanger sequencing of all the genes that are known to cause FH or through next-generation sequencing (NGS)-based custom packages.

## 5. Heterozygous FH

### 5.1. Cardiovascular Risk and cIMT as Cardiovascular Risk Marker

According to a retrospective cohort study using a primary care database of 1.5 million Spanish patients who were below 35 years of age, subjects with FH had a higher risk for ASCVD than subjects without FH. This was particularly true for those who had additional risk factors or had suffered from ASCVD in the past [31]. Based on the results of 12 different studies including 20,000 participants, Khera et al. (2016) demonstrated a 22-fold increased risk for CAD in patients with an LDL-C ≥ 190 mg/dL (4.9 mmol/L) plus an FH-causing variant when compared to individuals without an FH-causing variant and an LDL-C below 130 mg/dL ((3.4 mmol/L), OR 22.3, 95% CI 10.7–53.2) [32]. Measurement of the carotid intima-media thickness (cIMT) by ultrasound is a noninvasive, widely accepted, and reliable surrogate marker to monitor the development of atherosclerotic burden [33]. Children with HeFH (13.0 (range 8.0–18.0) years) showed significantly higher mean (SD) cIMT compared to their unaffected siblings (0.494 mm (0.051) versus 0.472 mm (0.049), *p* = 0.002) [34]. In a randomized, placebo-controlled trial, the treatment group (pravastatin 20 to 40 mg, during 2 years) experienced a 25% reduction in LDL-C levels and a substantial decrease in cIMT (mean (SD), −0.010 (0.048) mm; *p* = 0.049), whereas the placebo-treated group showed an increase in cIMT (mean (SD), +0.005 mm (0.044) mm; *p* = 0.28). Delta IMT was already significantly different between both cohorts within two years (*p* = 0.02) [35]. Treatment is required to reduce the cIMT of FH patients to levels of unaffected individuals. According to a systematic review and meta-analysis, published in 2022, based on approximately 6000 individuals, a smaller mean (95% CI) difference in cIMT was found between treated FH patients and controls, 0.05 (0.03–0.08) mm (*p* < 0.001), than between untreated FH patients and controls, 0.12 (0.03–0.21) mm (*p* = 0.009), and this supports the theory that early treatment is beneficial [36].

### 5.2. Conventional Treatment of Patients with HeFH

Pharmaceutical treatment is required if the LDL-C target for children with HeFH is not achieved with lifestyle recommendations. Currently, daily oral statins are the first step in pharmacological treatment [1]. Statins are HMG-CoA reductase inhibitors that prevent the synthesis of cholesterol and raise LDLR levels, which promote the absorption of LDL-C from the plasma. Statins are recommended in children ≥ 8 years old with LDL-C levels > 4.0 mmol/L. Addition of ezetimibe is recommended in children ≥ 10 years old when LDL-C is >3.5 mmol/L [1]. Ezetimibe impairs the absorption of fat in the intestine, and ezetimibe monotherapy results in a 27.0% lowering of LDL-C [37]. If LDL-C is still >3.5 mmol/L with the maximum dose of tolerated statins and ezetimibe, novel therapies need to be considered. Statin-associated muscle symptoms (SAMS) are an adverse effect that are most commonly mentioned as one of the primary causes of statin nonadherence and/or discontinuation in adults [38]. Although SAMS in children are not extensively described in the literature, it can be considered to substitute statins in children who experience adverse effects for another type of medication, such as ezetimibe or one of the novel treatment options that are discussed below. How to accurately diagnose SAMS is described extensively elsewhere [39].

### 5.3. Novel Therapies in HeFH

As described above in the pathophysiology section, gain-of-function variants in *PCSK9* are associated with hypercholesterolemia. Therefore, therapies that are focused on blocking PCSK9 result in a reduction in LDL-C levels. One of these PCSK9 inhibitors (PCSK9-i) is evolocumab, a monoclonal antibody that is administered via biweekly or monthly subcutaneous injections. A randomized, double-blind, placebo-controlled trial (NCT02392559) showed a significant absolute reduced LDL-C in children using evolocumab versus placebo (−2.0 vs. −0.2 mmol/L) (*p* < 0.001) [40]. Another PCSK9-i, alirocumab (also a monoclonal antibody), was also safe and effective in lowering LDL-C levels in children on background statin therapy (~45% reduction in LDL-C levels) (NCT02890992) [41]. Moreover, the newly developed inclisiran is a PCSK9-i but with a different mode of action, namely through gene silencing. Small interfering ribonucleic acid (siRNA) is administered via an intial subcutaneous injection, followed by a second dose at 3 months and, subsequently, every 6 months. This siRNA molecule prevents the translation of PCSK9-mRNA to PCSK9. Raal et al. (2020) showed a decrease of 39.7% (95% CI −43.7 to −35.7%) in LDL-C in adult patients who received inclisiran versus an increase of 8.2% (95% CI 4.3–12.2%) in LDL-C in patients who received placebo in a phase 3, double-blind randomized trial (NCT03397121) [42]. Inclisiran trials in children (12–17 years) are currently ongoing (ORION 16 and NCT04652726) [43]. Although more research is required to determine the long-term safety and effectiveness of PCSK9-i in pediatric patients, the addition of PCSK9 inhibition to background lipid-lowering therapy is promising.

Another promising therapy for children with FH is bempedoic acid. Bempedoic acid was approved for adult patients with HeFH by the U.S. Food and Drug Administration (FDA) and European Medicines Agency (EMA). This oral pro-drug is converted to bempoyl-CoA by the enzyme very-long-chain-acyl-CoA-synthetase 1 (ACSVL1) and inhibits the formation of Acetyl-CoA and, subsequently, the formation of HMG-CoA. As a result of the decrease in intrahepatic cholesterol, the number of LDLR increases and diminishes the LDL-C plasma with 16% in adult patients with HeFH, ASCVD, or both versus placebo [44]. As ACSVL1 is not present in muscle cells, muscle symptoms may be less common with bempedoic acid compared to statins. Therefore, bempedoic acid might be an interesting option for children with SAMS. As bempedoic acid is known to increase serum statin levels (potentially leading to an increased risk of myopathy), concomitant use with simvastatin (>20 mg) and pravastatin (>40 mg) must be avoided. Additionally, adult patients who use bempedoic acid have an increased risk of gout (incidence 1.2% versus placebo) and risk of tendon rupture (incidence 0.3% in OLE study) [45,46]. In children prone to SAMS, investigation is needed to explore if bempedoic acid in combination with (low-dose) statins prevents (worsening) myopathy. Results of a large trial on the effects of bempedoic acid on cardiovascular outcome in subjects with verified statin resistance and increased LDL-C levels are expected soon [47,48]. A trial on bempedoic acid in children will start in the near future.

## 6. Homozygous FH

### 6.1. Cardiovascular Risk and cCTA as Cardiovascular Marker

If left untreated, HoFH can result in fatal and nonfatal myocardial infarctions as early as in the first two decades of life [49]. Lifestyle optimization and maximum statin therapy (and other lipid-modifying therapies) do not result in a sufficient reduction in LDL-C in most subjects affected by HoFH. Due to the early onset of atherosclerotic plaques in individuals with HoFH, imaging is crucial for monitoring the presence and growth of atherosclerotic burden. The introduction of CT scanning with low-dose radiation has made coronary CT angiography (cCTA) more suitable for routine use in children [50]. Furthermore, a single-center cross-sectional study established that low-dose cCTA is more accurate than echocardiography for the early diagnosis of subclinical coronary and aortic root atherosclerosis in children with HoFH [51]. The current recommendation is to perform an annual echocardiogram and a low-dose cCTA every five years to evaluate the development of atherosclerosis in children with HoFH [18].

### 6.2. Conventional Treatment in Patients with HoFH

Since conventional therapy (statins and ezetimibe) generally does not reduce LDL-C to acceptable levels in patients affected by HoFH, lipoprotein apheresis, which physically removes LDL-C from the circulation, should be added. However, lipoprotein apheresis is not available in every country. King et al. published the first report on long-term plasmapheresis for the treatment of hypercholesterolemia in two children in 1980 [52]. Since then, the safety of this therapy has substantially improved and it is established as an effective therapy to acutely diminish the LDL-C levels in children with FH by up to 63–71% [53]. Since LDL-C levels increase again quite quickly due to endogenous LDL-C synthesis, lipoprotein apheresis procedures are usually performed (bi)weekly to maintain average LDL-C levels below target. The safety and efficacy of lipoprotein apheresis treatment for adults and pediatric patients with HoFH has not been evaluated in randomized controlled studies due to ethical considerations [54]. If apheresis or management with other lipid-lowering therapies is not possible, liver transplantation can be an option. Although liver transplantation lowers LDL-C levels by addressing the molecular defects of LDL clearance (−80%), the consensus panel on HoFH of the EAS noted that there are major disadvantages, such as surgical complications and the requirement for life-long immunosuppressive medication [18,55]. As the efficacy and safety of numerous pharmacological treatments have been proven in recent years and multiple trials are currently ongoing, liver transplantations might not be needed in the near future. However, as, in some regions, these new pharmacological treatments will not become available in the near future, liver transplantation might be a comprehensible treatment option in some patients.

### 6.3. Novel Therapies in HoFH

For younger children with HoFH, a drug that is used in adults with HoFH, lomitapide, is now being investigated. Lomitapide inhibits microsomal triglyceride transfer protein activity, which results in a decrease in very low-density lipoprotein (VLDL) secretion from the liver and decreased LDL-C production [56]. A recent (2021) multicenter, observational, retrospective, uncontrolled study including 75 patients with HoFH showed that use of lomitapide (mean dosage 20 mg per day) as add-on on a background lipid-lowering therapy decreased LDL-C by 60%. The use of lomitapide can dramatically reduce the frequency of apheresis: 36.8% of the patients even discontinued lipoprotein apheresis after initiation of lomitapide [57]. First results from a phase 3 trial of lomitapide in children (5–17 years) are promising: after 24 weeks, the mean reduction in baseline LDL-C levels was 54% (*p* < 0.0001, 95% CI −62 to −45%) [58]. Due to frequent, but treatable, gastrointestinal side effects and high annual costs (in 2023, approximately EUR 600,000 per patient per year if on a dosage of 40 mg per day in the Netherlands), cost-effectiveness analyses are needed.

Another type of medication that has been developed for severe lipid disorders but no longer used is antisense oligonucleotide therapy. Mipomersen is an antisense oligonucleotide (ASO) targeted against Apo(b) mRNA. Administration of mipomersen leads to reduced production of LDL-C, VLDL-C, and Lp(a). Although a recent meta-analysis by Asthaneh et al. showed a promising reduction in adults in LDL-C compared to placebo (mean difference −24.79%, 95% CI −30.15 to −19.43%), it is not used routinely due to safety concerns [59]. Elevated transaminases, hepatic steatosis, hepatotoxicity, and low platelet counts are frequently stated. Although the FDA approved mipomersen for adults as adjunct therapy to lipid-lowering therapy for HoFH in 2013, the EMA rejected it, as information about long-term safety is needed and other effective, tolerable therapy is available. If used, regularly monitoring liver enzymes and platelet counts is required.

PCSK9 inhibitors, such as evolocumab and alirocumab, can be used in children with HoFH with a defective/defective or a defective/null variant. PCSK9-i is not effective in receptor null/null patients because the effect of inhibition depends on the residual LDLR function [60,61]. Children with HoFH (mean age 12.4 years) on alirocumab in an open-label, single-arm, international phase 3 research showed a mean (SD) LDL-C reduction of 4.1% (36.0) at week 12, and, in 50% of the children (*n* = 9), a reduction of ≥15% LDL-C was shown after 12 weeks, without any unexpected safety/tolerability findings [62]. PCSK9-i evolocumab showed a mean reduction in LDL-C of 21.2% in 12 weeks in an open, single-arm study of children with HoFH [63]. Inclisiran might be promising for HoFH children with at least one defective variant as well. Trials in adults (ORION 5 and NCT03851705) and in children (ORION 13 and NCT04659863) with HoFH are being conducted to evaluate the safety, tolerability, and efficacy of inclisiran on background standard therapy.

For those with HoFH and very little LDLR residual activity, new therapies that act through mechanisms that do not involve the LDL-C receptor are most promising. Evinacumab, a human monoclonal antibody that acts as an ANGPTL3 inhibitor, showed very promising results in clinical trials. After 6 months of treatment with evinacumab (15 mg/kg intravenously monthly), the LDL-C levels were dramatically lowered in two children with HoFH (12 and 16 years old, respectively), and a regression in total volume of atherosclerotic plaque, as evaluated by cCTA, of 76% and 85%, respectively [64], was shown. This suggests that, when appropriate treatment is received on time, atherosclerosis may be reversible in people with HoFH comparably to patients with HeFH, as long as a plaque is not calcified. Unpublished preliminary data suggest that the frequency of apheresis in children with HoFH could dramatically be reduced with a monthly infusion of evinacumab (manuscript in preparation, NCT04233918). Evinacumab is approved by the FDA and EMA for adults and children with HoFH from 12 years old. At this moment, evinacumab is being considered in children with HoFH from 5 years old by the FDA. However, due to high annual costs, cost-effectiveness analyses are needed. An overview of all treatment options for children with FH is shown in Figure 2.

## 7. Lp(a) in Children with FH

Elevated Lp(a) levels are a known risk factor for the development of cardiovascular diseases. Additionally, the onset of aortic valve stenosis in adults is closely correlated with a higher Lp(a) [65]. Lp(a) is a particle that resembles LDL-C but contains a ‘tail’ of apolipoprotein (a) (apo(a)). This apo(a) protein consists of many kringle-shaped loop structures. One of these kringles, namely kringle IV, has 10 subtypes. The second subtype (KIV_2_) can vary in the number of repeats. A lower number of repeats results in a shorter protein tail and higher Lp(a) levels, possibly due to slower synthesis of larger proteins. The number of repetitions is controlled by the LPA gene; polymorphisms cause the number of kringle IV repeats to vary between individuals from 10 repeats per Lp(a) to >50 repeats [66]. Lp(a) values are mostly determined by a person’s genetic background [67]. For a long time, it was hypothesized that Lp(a) values remain constant in an individual, regardless of age or lifestyle. However, more recently, it was shown that Lp(a) values are not steady during childhood; they increase rapidly after birth and are thought to start to level out around the first or second year of life. According to a study involving approximately 3000 children who visited a lipid clinic in the Netherlands, the mean increase in Lp(a) from 8 years to adulthood was 22% in individuals who used no medication, 43% in individuals on statins, and 9% in individuals on statins + ezetimibe [68]. These findings suggest that repeating the Lp(a) measurement at several points in time may be appropriate in some circumstances, particularly when levels are at the upper limit of normal, or during statin use. Whether an increased Lp(a) causes early atherosclerosis in children is still unknown. The cholesterol in Lp(a) can interfere with the measurement of LDL-C and falsely increase the LDL-C [69]. Therefore, all children suspected of having dyslipidemia should have an Lp(a) measurement performed to distinguish between elevated Lp(a) and FH. Additionally, it is also possible to find children who have both risk factors and may consequently be at a higher risk of early atherosclerosis [70]. It is unclear what levels of Lp(a) should be considered as high, but most (international) standards mention an elevated Lp(a) at values above 125 nmol/L (≈50 mg/dL). In order to reduce cardiovascular risk, it is crucial to keep all cardiovascular risk factors in children with increased Lp(a) as ideal as possible. A balanced diet, blood pressure management, diabetes prevention/management, no smoking, maintaining a healthy weight, getting enough exercise, and optimizing LDL-C levels below the treatment limit (3.5 mmol/L) are all important pieces of advice [70,71]. At this moment, no medication for children is available to reduce Lp(a) levels [70]. Although lipid apheresis can reduce Lp(a) up to 70%, this invasive treatment option is not widely used for individuals with high Lp(a) levels [72,73]. While studies on reducing Lp(a) with ASOs and siRNA are promising, long-term prospective studies about the efficacy and safety in children are needed. These therapies might be particularly important for children who have an elevated Lp(a) and another cardiovascular risk factor, such as FH [70].

## 8. Conclusions

Early screening, adequate treatment, and imaging for subclinical atherosclerosis are cornerstones in the management of children with FH. Due to the discovery of a ‘longevity effect’ of loss-of-function variants in *PCSK9* and *ANGPTL3* genes, which both lead to extremely low LDL-C levels, novel types of medication have been developed in the last few years. These new types of drugs specifically inhibit translation of these genes, which, in turn, drastically lower LDL-C levels and thereby minimize atherosclerotique burden. For children that cannot tolerate (due to SAMS) the necessary dosage of statins to reach their LDL-C targets, these new types of drugs are a valuable new treatment option. For children with HoFH, the frequency of invasive apheresis may dramatically be reduced and, therefore, can dramatically increase the quality of life in these patients. Lp(a) measurement is needed in children with dyslipidemia. Although no treatment for children with an elevated Lp(a) has been investigated at this moment, effective treatment in lowering Lp(a) may become available for adults in the near future.

## Figures and Tables

**Figure 1 genes-14-00669-f001:**
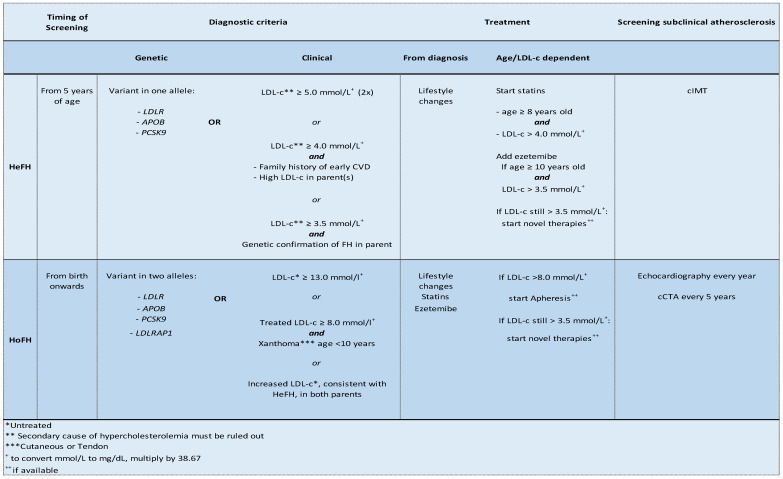
Overview of key aspects of timing of screening, diagnostic criteria (genetic and clinical), treatment, and options for screening for subclinical atherosclerosis as stated by the EAS. Children suspected for HeFH or HoFH need to be screened from 5 years of age and from birth onwards, respectively. When genetic confirmation is not possible, clinical diagnostic criteria can be used based on LDL-C levels, family history, or patient’s characteristics. In both HeFH and HoFH, lifestyle changes are needed from diagnosis, while the start and type of treatment differ between HeFH and HoFH and depend on age and LDL-C levels. Screening for subclinical atherosclerosis is possible in HeFH (in research setting) but recommended in HoFH.

**Figure 2 genes-14-00669-f002:**
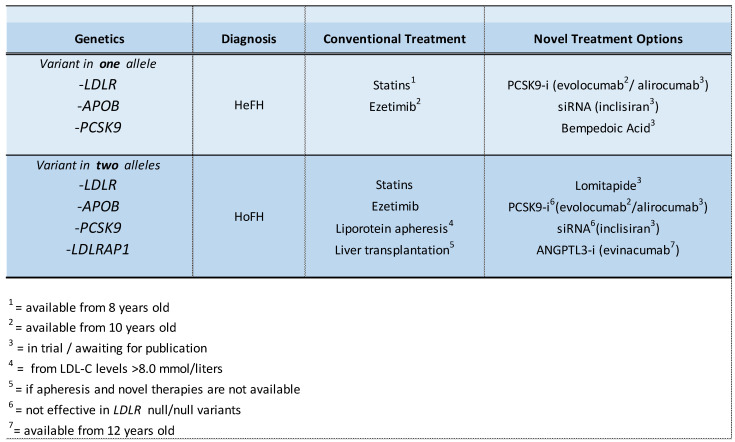
Genetic diagnosis and conventional/novel treatment options for children with FH.

## Data Availability

Data sharing not applicable.

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
