# Peer review of "How Genetic Variants in Children with Familial Hypercholesterolemia Not Only Guide Detection, but Also Treatment"

_genes, 2023, doi:10.3390/genes14030669_

Round 1
Reviewer 1 Report
This is a very comprehensive review on paediatric FH by the group of leading experts in the field, which add several important notions on this very important topic.
Few minor aspects could be improved:
- Title - this is very important notion, but it could be even better related to the manuscript content. Could you possibly add a comprehensive figure on how the genetic diagnosis could guide the diagnosis and especially treatment. In addition, the title is referring to paediatric population, but in the text and in the literature used, the delineation between adult and paediatric populations are not always clear or references are not referring to paediatric population.
- Some of the abbreviations are not consistently used after introduction or are re-introduced (e.g. SAMS)
- Some statements lack appropriate (more updated and/or more accurate) reference and/or are not very precisely written:
Page 2, lines 56-62: More recent references on the current global situation would be 10.1016/S0140-6736(21)01122-3;10.1016/j.atherosclerosis.2018.08.051
Page 2, lines 67-70: Please add the following references that reflect recent important developments on this field : 10.1093/eurheartj/ehac224;10.1093/eurjpc/zwac200; 10.3205/hta000136
Page 2, line 71: Recent global data 10.1016/S0140-6736(21)01122-3
Page 4, lines 99-110: Some populations could differ quite substantially from these figures: 10.1016/j.gim.2022.06.010
Page 5, lines 183-185: No reference is added. In addition, SAMS is children is less of a problem, the reference 35 is reffering to adults?
Page 7, lines 264-268: Liver Tx might still be good option in some settings (lack of access to therapies); immunosupression might not need to be life-long; some long-term reports are quite promissing - for the more nuanced review please see the review part of the 10.3389/fped.2020.567895
- In addition, some references seem a bit outdated or redundant or more appropriate ones could be used (e.g. 12, 15, 27, 35, 47, 61, ...).
- The Conclusion could be improved to more precisely reflect the main messages of the manuscript.
Reviewer 2 Report
I have attached recommendations additions and multiple grammatical changes

Reviewer 3 Report
The manuscript of van den Bosch et al. is a comprehensive review on the topic of Familial Hypercholesterolemia. The authors highlight the relatively high frequency of the disease and therapeutic possibilities.
The manuscript has a clear structure and encompasses screening, pathophysiology and genetics, diagnosis and treatment. Clinical recommendations as well as recommendations regarding further diagnostics are incorporated. The references are up to date. Figure 1 helps to understand the screening process and its timing. Conclusions are based on the manuscript content.
Minor comments:
1. How does a molecular diagnosis look like (methods)? What is cost-effectiveness?
2. Recently, there is a lot of interest in RNA therapies. Maybe it would be an idea to mention it here.
Round 2
Reviewer 2 Report
This paper still needs editing for English grammar. I extensively edited previously and do not have the time to do it again. I trust Genes will do this.
Lines 297-298 Apheresis acutely diminishes LDL (there is a rapid rebound so it is important to point this out)
Amryt just announced positive results in HoFH children ages 5-17 in terms of LDL reduction with lomitapide. This should be included.
Evinacumab is being considered for use in HoFH children ages 5 and up by FDA.
last line of the paper. It is not certain that Lp(a) lowering medications will become available - should say may become available
